# Applying Membrane Distillation for the Recovery of Nitrate from Saline Water Using PVDF Membranes Modified as Superhydrophobic Membranes

**DOI:** 10.3390/polym12122774

**Published:** 2020-11-24

**Authors:** Fatemeh Ebrahimi, Yasin Orooji, Amir Razmjou

**Affiliations:** 1College of Materials Science and Engineering, Nanjing Forestry University, Nanjing 210037, China; rto.est@gmail.com; 2Department of Nanotechnology, Faculty of Advanced Sciences and Technologies, University of Isfahan, Isfahan 73441-81746, Iran; 3Department of Biotechnology, Faculty of Advanced Sciences and Technologies, University of Isfahan, Isfahan 73441-81746, Iran; 4UNESCO Centre for Membrane Science and Technology, School of Chemical Science and Engineering, University of New South Wales, Sydney 2052, Australia

**Keywords:** nanocomposite membrane, membrane distillation, nitrate, sustainable development, simulation

## Abstract

In this study, a flat sheet direct contact membrane distillation (DCMD) module was designed to eliminate nitrate from water. A polyvinylidene fluoride (PVDF) membrane was used in a DCMD process at an ambient pressure and at a temperature lower than the boiling point of water. The electrical conductivity of the feed containing nitrate increased, while the electrical conductivity of the permeate remained constant during the entire process. The results indicated that the nitrate ions failed to pass through the membrane and their concentration in the feed increased as pure water passed through the membrane. Consequently, the membrane was modified using TiO_2_ nanoparticles to make a hierarchical surface with multi-layer roughness on the micro/nanoscales. Furthermore, 1H,1H,2H,2H-Perfluorododecyltrichlorosilane (FTCS) was added to the modified surface to change its hydrophobic properties into superhydrophobic properties and to improve its performance. The results for both membranes were compared and reported on a pilot scale using MATLAB. In the experimental scale (a membrane surface area of 0.0014 m^2^, temperature of 77 °C, nitrate concentration of 0.9 g/Kg, and flow rate of 0.0032 Kg/s), the flux was 2.3 Kgm^−2^h^−1^. The simulation results of MATLAB using these data showed that for the removal of nitrate (with a concentration of 35 g/Kg) from the intake feed with a flow rate of 1 Kg/s and flux of 0.96 Kgm^−2^h^−1^, a membrane surface area of 0.5 m^2^ was needed.

## 1. Introduction

The vast variety of nanotechnology applications is triggering a breakthrough across many fields [1,2,3,4,5,6,7]. Water and water resources are among the fundamental pillars of sustainable development in any country [8,9,10]. Thus, they should be preserved and managed in such a way that the needs of future generations for the supply of this essential material are taken into account [11]. The shortage of freshwater supplies on the earth and climate change leading to recent droughts in some countries have turned water scarcity into a global and environmental concern [12]. This shortage entails the necessity of paying more attention to preserving the existing resources and recycling wastewaters [12]. In this regard, the quantitative and qualitative preservations of water resources especially in dry and waterless areas as well as soil preservation are of great importance in the survival of the earth and the living organisms on it [13,14,15]. Furthermore, the infiltration of different types of pollutants as well as urban, industrial, and agricultural wastewaters into the groundwater have had long-term adverse effects on the quality of water resources and people’s health [16,17,18]. Chemical materials play a significant role in the contamination of the groundwater and wastewater. A considerable number of these pollutants (especially nitrate) originate from different types of fertilizers used in agricultural activities [19,20].

The consumption of water polluted with nitrate is harmful to human health (especially to children) and sometimes causes severe harms and even death. Gastrointestinal bacteria in the stomach have the ability to transform nitrate to nitrite. A high amount of nitrite in the body can lead to methemoglobin disease, infant death, and abortion. Nitrite can also be combined with amines or amides in the body and form nitrosamine, a well-known carcinogen [21,22]. Hence, nitrate removal from drinking water is of great importance from environmental and health perspectives. Although water pollution with nitrate has several reasons, the use of nitrate fertilizers and pesticides in agriculture is considered as one of the most important reasons [23,24,25]. According to the World Health Organization (WHO), the permitted nitrate level in drinking water is 50 mg/L (in nitrate). Furthermore, the United States Environmental Protection Agency has announced the maximum allowed level as 10 mg/L in nitrogen equivalent to 44.82 mg/L in nitrate [26]. Nitrate can also impose serious environment issues. Over-supply of nitrate-based fertilizers results in a contamination of soil which makes the farmlands unsuitable for future farming. In addition, the high concentration of nitrate in water resources causes the reduction of its oxygen level damaging aquatic life.

Membrane distillation is a relatively new method for purifying solutions and treating water. This method which is a combination of distillation and membrane separation methods not only possesses the advantages of both of these methods but also lacks their disadvantages to a large extent. The basis of separation in both distillation and membrane distillation processes is the liquid–vapor equilibrium. Moreover, they both change their phases by receiving the latent evaporation heat. In conventional distillation, it is necessary to heat the feed to the boiling point, while the membrane distillation process is carried out at a temperature below the boiling point; therefore, energy is saved. Membrane distillation is a separation method based on using a membrane and has solved many problems of other membrane separation methods, such as polarization and increased concentration, high energy requirements, and the need for multiple purification steps.

The driving force of membrane distillation is the temperature difference between the two sides of the membrane which leads to a difference in the vapor pressure on its two sides and the passage of water vapor through it. Water cannot move to the other side because the membrane is hydrophobic or superhydrophobic. Additionally, the surface tension force of water and the repulsive force between the water droplets and the hydrophobic membrane used in this process prevent water droplets from passing through the membrane pores. However, water vapor can easily pass through them. In other words, instead of using the distillation column, the pore volume of the porous membrane is used which reduces the costs and saves space and equipment. Since this process is based on the passage of vapor, non-evaporable ions, colloids, and macromolecules cannot pass through the membrane and return to the solution. Additionally, other materials with a boiling point higher than the feed temperature cannot pass through the membrane. As a result, the permeate is highly pure. Thus, in this process, water is separated without being wasted and is independent of the feed concentration [27]. The hot feed and the cold pure water flow on the two sides of the membrane in opposite directions. Another advantage of the counter-current module is that the chance of membrane fouling by solid materials in this method is less than that of other methods. As a result, the membrane becomes more durable [28]. Due to the rotational water flow in this process, the residual thermal energy of the feed is not wasted and returns to the system.

The main advantage of this process is that it is done at atmospheric pressure at a much lower temperature than the boiling point of water. This advantage has led to widespread applications for this method. Meanwhile, to supply this low temperature, renewable energy sources including geothermal and solar energies may be employed at household or industrial scales. Direct Contact Membrane Distillation (DCMD), Sweeping Gas Membrane Distillation (SGMD), Air Gap Membrane Distillation (AGMD), and Vacuum Membrane Distillation (VMD) are four MD configurations.

Razmjou et al. (2012) reported that they managed to make a PVDF membrane superhydrophobic in order to apply it in DCMD membrane distillation [29]. They found in their study that pore wetting was a major problem in membrane distillation. When the pores get wet, the feed partially or entirely passes through the pores, contaminates the permeate, and blocks the pores [30]. Hence, the cross-membrane flux and the membrane performance are reduced. To solve this problem, they engineered the surface of the hydrophobic membrane and converted it into a superhydrophobic one. This modification significantly reduced the direct contact surface between the membrane and the feed which reduced the wettability of the pores, their contamination, heat loss near the membrane, and also the driving force required by the process. Boubakri et al. (2013) removed nitrate from water with a purity of 99.90% by direct membrane distillation and using polypropylene (PP) and polyvinylidene fluoride (PVDF) hydrophobic membranes [27]. Their results showed that the limitations of the methods mentioned above including the relationship between nitrate removal and anions with nitrate and nitrate concentration did not affect the results of the process.

Zhang et al. (2013) made a superhydrophobic PVDF membrane to apply it in DCMD [31]. They developed a superhydrophobic composite membrane by coating a mixture of PolyDiMethylSiloxane (PDMS) and SiO2 hydrophobic nanoparticles on a PVDF membrane so that the contact angle changed from 107 to 156 degrees. Although the cross-membrane water flux of the modified membrane was reduced compared to that of virgin membrane, the permeate was purified of sodium chloride with a purification of 99.99%. In addition, the modified membrane showed significant anti-fouling properties. Dong et al. (2016) simulated the removal of NaCl from water by membrane distillation technology [18] on the industrial scale.

The aim of this research was to optimize the hydrophobic properties of the membrane in two stages: first, changing the topography of the surface to enhance roughness by coating TiO_2_ nanoparticles on the surface of the membrane; second, changing the chemical properties of the surface to repel water more using the functional agent FTCS. The modified membrane was used in a DCMD membrane module in the experimental scale. Then the results of this optimization were evaluated by using Dong’s MATLAB codes for NaCl [18]. Because water polluted with NaCl and water polluted with nitrate are similar in terms of physical properties, such as density and the number of ions, these codes were used for the feed containing nitrate. Then, the obtained experimental results were given to MATLAB software which changed them into the semi-industrial scale.

## 2. Materials and Methods

### 2.1. Materials

In this study, the following materials (provided from Sigma-Aldrich) were used: polyvinylidene fluoride (PVDF) flat-sheet membrane HVHP (Millipore, nominal pore size: 0.45 µm, porosity: 75%), titanium (IV) isopropoxide (TTIP) (98%) as TiO_2_ precursor, 2,4-pentanedione, acetylacetone, Milli-Q Water, perchloric acid (70%), 1H,1H,2H,2H-perfluorododecyltrichlorosilane (FTCS), potassium nitrate, potassium iodide, absolute ethanol, and toluene.

### 2.2. Membrane Surface Modification

In order to synthesize titanium oxide nanoparticles (Figure 1), ethanol, 2,4-pentanedione, perchloric acid 70%, titanium tetraisopropoxide, and Milli-Q water were mixed, at room temperature to form a stable sol of the nanoparticles. The sol was stirred for an hour. The molar ratios of each component in the resulting sol were TTIP: Pluronic F127: 2,4-pentanedione: HClO_4_:H_2_O: Ethanol = 1:0.004:0.5:0.5:0.45:4.76. PEG (1000 g/mol) is also substituted with the Pluronic F127 to see the effect of hydrophilic templating agent on the coating films [32].

The operation was carried out by the dip-coating process. The stages of this process are explained below (Figure 2). The dip coater was used to coat TiO_2_ nanoparticles on the membrane by dipping the membrane in the sol-gel solution at a speed of 50 mm/s and holding it in the solution for 8 s. Then, the membrane was removed at the same speed. Afterward, the sample was dried in an oven at 120 °C for one hour. Some nanoparticles do not create a strong chemical bond during the coating process or merely create a weak physical bond with the surface. The hydrothermal process is performed to increase the energy of these particles to make a strong bond with the surface after coating the nanoparticles on the membrane surface. The membrane coated with a sol-gel solution was placed in a sealed container overflowed with Milli-Q water and was placed in an autoclave at 90 °C for 2 h. After coating and heat treatment, the sample was radiated with ultraviolet light (UV) for 6 h. The residual organic sediments were decomposed by this radiation. Finally, the membrane was called ‘TiO_2_-PVDF’.

### 2.3. Fluorination of the Surface of the TiO_2_–PVDF Membrane

First toluene was purged with nitrogen for one hour to remove its oxygen content as much as possible. Then, to reduce its temperature, toluene was refrigerated for 2 h. FTCS powder was then added to toluene within a controlled temperature in the range of 0 to −5 °C. Then while the solution was stirred it was treated with an ice bath. The FTCS solution was coated on the membrane surface of the TiO_2_–PVDF nanocomposite by the self-assembly method. In this method, after preparing the FTCS solution, the samples coated with titanium nanoparticles were placed in the solution so that the FTCS molecules were bonded with the TiO_2_–PVDF membrane by the self-assembly method (Figure 3). Hereafter, the membrane is called the ‘FTCS–TiO_2_–PVDF’ membrane.

### 2.4. The Performance of the Membranes with Synthetic Feed Waters

After ensuring the good performance of the module by conducting the hot/cold water test, the main test was conducted at the atmospheric pressure of one for 6 h in two phases with direct contact membrane distillation setup (Figure 4 and Appendix A). The first phase included operations using an unmodified membrane and the second phase involved operations with a modified membrane. The temperature difference of the feed and the permeate was kept constant during the process (about 50 degrees on average). First, water containing nitrate was synthesized by adding 5 wt % potassium nitrate.

Parallel counter-currents (the hot-water feed containing nitrate ions and the cold water containing pure permeate) were formed using two peristaltic pumps with a constant speed of 37 rpm. The feed container was placed in a hot water tank whose temperature was controlled. The permeate container was placed in a tank containing a mixture of water and ice and its temperature was controlled. The temperatures of the feed and permeate were controlled by isolating the system. The water vapor passed through the membrane from the feed side to the permeate side. In the cold side, vapor lost its energy and became water. During the test, the weight and volume of the feed gradually decreased, while the volume and weight of the permeate increased. The amount of ion in the feed and permeate containers was measured by an electrical conductivity meter every hour. The weights of the feed and permeate were also measured by a scale every hour.

### 2.5. Pilot Process Simulation

Dong et al. [33] created a mathematical approach by pairing tanks-in-series and a black box to investigate all the main factors of the DCMD process versus the length of the membrane. The designed simulator was used to predict and evaluate the performance of the flat-sheet DCMD at an industrial scale, as well as the impact of the physical characteristics of the membrane, module dimensions, and the implementation of conditions on the performance of the large-scale DCMD module. This research was conducted to obtain the design considerations for the production of pure water. In their study, the software required operational and laboratory-scale data to produce the desired outputs in large-scale countercurrents (Appendix A). In the current research, the experimental conditions of Dong et al.’s study were reconstructed. However, nitrate was used instead of NaCl. Due to the similarities between the physical properties of nitrate and NaCl, Dong’s MATLAB codes were used to convert experimental results to large-scale results.

## 3. Results and Discussion

### 3.1. Surface Morphology

The morphology of membranes was characterized by FE-SEM (QUANTA FEG 450, Hillsboro, OR, USA) operating at 15 kV. The comparison of several scanning electron microscopy (SEM) images of the FTCS–TiO2–PVDF membrane and the PVDF membrane (Figure 5a,b) shows that a multi-level hierarchical roughness was created on the surface of the membrane at the end of the process which prevented water droplets from penetrating the surface. Furthermore, while the contact angle increased substantially, the average pore size did not change significantly (Figure 5c–f). It was expected that the performance of the modified membrane was better than that of the unmodified membrane. Furthermore, the microscopic changes on the surface of the membrane were investigated in different stages.

As can be seen in (Figure 5g,h), coating TiO_2_ nanoparticles did not significantly change the porosity, pore size, and surface structure of the membrane. These nanoparticles are important in the creation of superhydrophobic properties in two regards: first, they increase the surface roughness; second, they are an anchor for linking the FTCS molecules on the surface.

The microscopic images of coating FTCS on the membrane (Figure 5i,j) show that the self-assembly method was able to create a more uniform coating on the surface than previously reported methods. As can be seen in this figure, despite the penetration of FTCS into the pores, no significant surface structure changes occurred, and the size of the pores did not change. It should also be noted that preserving the size and structure of the pores is crucial in the performance of DCMD. FTCS was used to reduce surface energy chemically. However, as can be observed in (Figure 5i,j) FTCS also increased hydrophobicity by creating roughness. FTCS crystals increased the surface roughness (Figure 5k,i). Furthermore, the images show that the sizes of the pores did not change substantially (Figure 5i–q) SEM images of different samples of the FTCS–PVDF membrane in different magnifications (Figure 5m–q).

In a study carried out by Razmjou et al. [29], the increase of the contact angle in the stage of coating FTCS on PVDF was reported as 146° ± 5° (the contact angle for the virgin membrane was 125° ± 1°) while in the current work which was done by the self-assembly method, the contact angle was 142° ± 20° (the contact angle for the virgin membrane was 89° ± 8°). The contact angles formed by water droplet on the membranes surface were measured by the sessile drop technique (CA-500A instrument, Yasin Pajooh Co. Ltd., Isfahan, Iran).

### 3.2. Investigating the Durability and Photocatalytic Activity of TiO_2_ Nanoparticles Coated on the Surface

The KI test was employed to ascertain the durability of TiO_2_ nanoparticles on the surface of the PVDF membrane (Figure 6). If the coated membrane has a photocatalytic activity, it can oxidize ion (I^−^) according to Equation (1) and convert it to (I_2_) and change the color of the colorless solution KI to yellow color.
(1)$$I^{-}+h^{+}\to ½\;I_{2}\;I_{2}+I^{-}\to I_{3}^{-}$$

The color change qualitatively indicates the photocatalytic activity of the sample coated by TiO_2_ nanoparticles. The UV-Vis absorption spectroscopy device was used for the quantitative evaluation of the photocatalytic properties of the sample. The UV-Vis spectrophotometer showed the highest amount of absorption at the wavelengths of 288 and 351 nm which represent the presence of I_2_ and $I_{3}^{-}$ atoms, respectively. To check the durability of TiO_2_ nanoparticles, the KI test may be done under UV radiation in different periods. In the solution containing the unmodified membrane, no change of color and no absorption occurred after 6 h of UV radiation, while in the solution containing the membrane coated with TiO_2_ nanoparticles both color change to yellow and absorption peak were observed. As is shown in Figure 7 absorption in each period for both samples remained relatively unchanged. An unchanged absorption spectrum during the period not only reflects its excellent photocatalytic activity, but is also proof of the suitable durability of TiO_2_ coatings.

### 3.3. Membrane Hydrophobicity

As was mentioned above, in this research, the hydrophobic properties of the membrane were optimized in two stages: first, changing the topography of the surface to enhance roughness by coating TiO_2_ nanoparticles on the surface of the membrane; second, changing the chemical properties of the surface to repel water more using the functional agent FTCS. Then the modified membrane was used in a DCMD membrane module at the experimental scale.

Although the coating of TiO_2_ nanoparticles on the surface of the PVDF membrane can reduce its hydrophobic properties due to the hydrophilic properties of TiO_2_, in this study, the purpose of using titanium oxide nanoparticles was to create roughness and the hydrophilic modification of the membrane was not the aim. A multi-layer roughness can change the wettability properties of a membrane [34,35]. An increase in surface roughness (after the surface is coated with TiO_2_ nanoparticles and a multi-layer roughness is formed on it) can lead to a strong capillary water suction on the membrane surface which expands the droplet on the surface until saturation [36]. Therefore, the hydrophilic properties of the surface increase.

Then the PVDF membrane was coated by FTCS and a substantial reduction was observed in the hysteresis angle from 18° for the unmodified membrane to about 6° for the FTCS–PVDF membrane. The increase in the water contact angle for the FTCS–PVDF membrane surface may be because of the notable reduction of the free energy of the membrane surface and its increased roughness.

Hydrophobic surfaces repel water and an air gap is created between the membrane and water droplets. This gap decreases the wettability of the membrane surface. In this case, water droplets can only have contact with the tip of the roughness; therefore, the physical contact between water droplets and the surface is considerably reduced. This is due to the fact that an air gap is created among the roughness bumps and surface tensile forces. Water droplets on superhydrophobic surfaces have small hysteresis contact angles and lie on the membrane surface spherically with a good approximation. Thus, they can roll even on surfaces with low slopes and remove pollutants from the surface (called the ‘self-cleaning’ property).

To increase the hydrophobic properties of the membrane, the TiO_2_–PVDF membrane was coated by FTCS. The unmodified PVDF membrane had the contact angle of 89° ± 8° degrees with water. It is worth noting that the surface modification (FTCS–TiO_2_–PVDF) created a contact angle of about 174° ± 10°. The unmodified and TiO_2_–PVDF membranes lacked self-cleaning properties on steep slopes, while the FTCS–TiO_2_–PVDF membrane showed self-cleaning properties even at low slopes.

### 3.4. Membrane Distillation at the Experimental Scale

Tests with a 5 wt % potassium nitrate solution were conducted for the unmodified and modified membranes. When the unmodified membrane was used, the cross-membrane water flux was 2 × 10^−3^ Kg/m^2^s and when the FTCS–TiO_2_–PVDF membrane was employed, the cross-membrane water flux was 6.4 × 10^−4^ Kg/m^2^s. When each of the membranes was used, the electrical conductivity of the permeate was zero. In general, flux reduction was observed in both cases over time which could have two reasons. The first reason may be the blockage or fouling of the membrane pores and the reduced surface area available for the passage of water vapor; the second reason could be the decrease of the driving force (i.e., the reduction of temperature difference between the feed and permeate) [37]. To remove the fouling of the pores, each membrane was placed for 15 min in a NaOH 0.2 wt % solution with the pH of 12 at room temperature (25 °C). After being washed and reused, the modified membrane had the recovered flux of up to 95% and the unmodified membrane had the recovered flux of about 60%. After the membranes were washed, colors and sediments were removed from the modified membrane, while in the unmodified membrane, the color change was not resolved requiring more time for washing and sometimes backwashing. This phenomenon can show the anti-sedimentation property of the modified membrane. In the current research, the coat is easy to produce due to the low production temperature and the easy methods selected for coating. Hence, in case of damage to the coat of the membrane during washing, the coat can be easily restored.

### 3.5. Physical and Chemical Characteristics of the Permeate

As can be observed in Figure 8a, the electrical conductivity of the feed had an ascending trend and increased from 200 to 244 μs during a 6 h continuous test. Nevertheless, the electrical conductivity of the permeate remained constant throughout the test and was equal to that of distilled water. This result indicates that no ion was able to penetrate to the other side of the membrane and the permeate was pure.

Moreover, the mass of the feed had a descending trend and the mass of the permeate had an ascending trend indicating the transfer of mass through the membrane from the feed side to the permeate side Figure 8b.

In what follows, the above experiment was conducted using the FTCS–TiO_2_–PVDF membrane. As can be seen in Figure 8c, during 6 h of continuous testing, the electrical conductivity of the feed had an ascending trend and increased from 200 to 208.6 μs, while the electrical conductivity of the permeate remained constant during the experiment and was equal to that of distilled water.

Moreover, the mass of the feed had a descending trend and the mass of the permeate had an ascending trend indicating the transfer of mass through the membrane from the feed side to the permeate side Figure 8d.

Comparing Figure 8a,c shows that the slope of the feed of the PVDF membrane is higher than that of the FTCS–TiO_2_–PVDF membrane which may be due to the higher flux in the PVDF membrane. Interestingly, the slope of the permeate is almost constant in Figure 8b,d, indicating the purity of the permeate.

The interesting point about the performance of the unmodified and modified membranes in the two tests mentioned above is the difference between the increased electrical conductivity of the feed in them over 6 h. According to the mass-time diagrams, this difference could be due to the lower mass reduction of the modified membrane Figure 8b than that of the unmodified membrane Figure 8d. There are detailed explanations regarding mass transfer reduction. As can be observed in Figure 8d, the flux is not as expected which is a disadvantage of this system and needs to be improved. Future studies may focus on solving this problem.

### 3.6. Process Simulation Using MATLAB at Pilot Scale

At this point, using the MATLAB software [33], the obtained results for the modified membrane at the laboratory scale were simulated at the pilot scale as shown in Table 1.

What is noticeable in the simulation is the low temperature of the feed (below 50 °C) at the pilot scale. Another point to note is the ability of this method to remove high concentrations of nitrate at the pilot scale. According to the data obtained from the simulation, a small area of the modified membrane (0.5 m^2^) is enough to remove nitrate from water with the concentration of 35 g/kg in a low temperature (48.33 °C) with the flow rate of 1 Kg/s and cross-membrane flux (0.96 Kgm^−2^h^−1^). The reason for the reduction of flux at the pilot scale may be the decrease of the driving force temperature difference). With the increase of temperature difference, the flux will probably increase (Appendix A).

## 4. Conclusions

The PVDF membrane was coated in two steps with TiO_2_ nanoparticles and 1H,1H,2H,2H-Perfluorododecyltrichlorosilane. The hierarchical structure of the membrane with a multilayer roughness resulted in an increase in the contact angle of the membrane from 89° ± 8° to 174° ± 10°. The membrane performance was evaluated by the direct contact membrane distillation method using 5 wt % potassium nitrate at the laboratory scale. The electrical conductivity of the feed containing nitrate increased, while the electrical conductivity of the permeate remained constant during the entire process. Furthermore, the low temperature and high flux at the simulated pilot-scale indicate that this method is highly efficient.

At the pilot-scale, using the membrane distillation method and 0.5 m^2^ of the modified membrane, it is possible to treat water polluted with nitrate with the concentration of 35 g/Kg at the low temperature of 48.33 °C and the flow rate of 1 Kg/s with the cross-membrane flux (0.96 Kgm^−2^h^−1^).

## Figures and Tables

**Figure 1 polymers-12-02774-f001:**
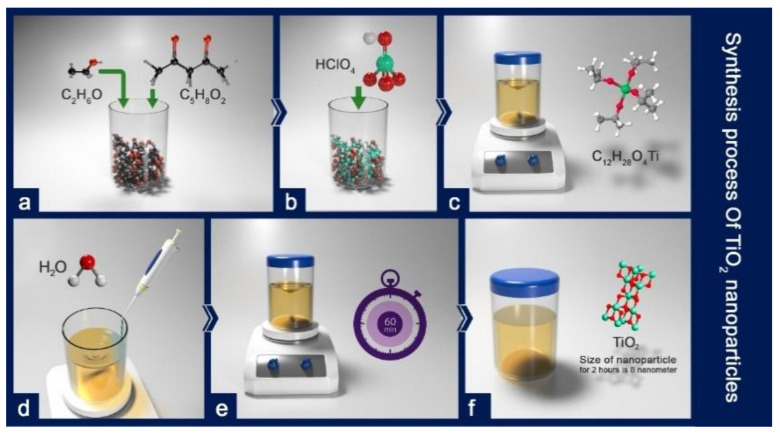
The synthesis of TiO_2_ nanoparticles using the sol-gel method: (**a**) mixing ethanol and 2,4-pentanedione; (**b**) adding perchloric acid 70%; (**c**) stirring the solution and adding titanium tetraisopropoxide; (**d**) adding Milli-Q water dropwise; (**e**) stirring the solution for an hour at room temperature; (**f**) the TiO_2_ sol gel solution.

**Figure 2 polymers-12-02774-f002:**
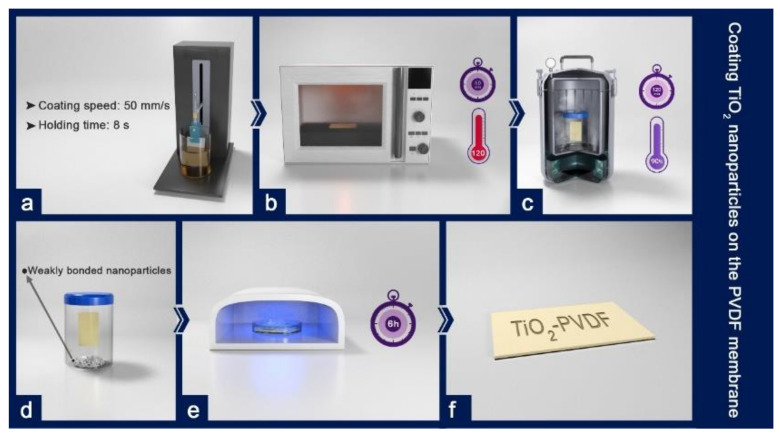
Coating TiO_2_ nanoparticles on the PVDF membrane: (**a**) dip-coating the membrane in the TiO_2_ sol-gel solution (coating speed: 50 mm/s; holding time: 8 s); (**b**) drying the membrane at 120 °C for an hour; (**c**) treating the membrane using the hydrothermal method at a low temperature (90 °C) for two hours; (**d**) the separation of the weakly bonded nanoparticles after heat treatment; (**e**) exposing the membrane to UV light for six hours; (**f**) the TiO_2_–PVDF membrane.

**Figure 3 polymers-12-02774-f003:**
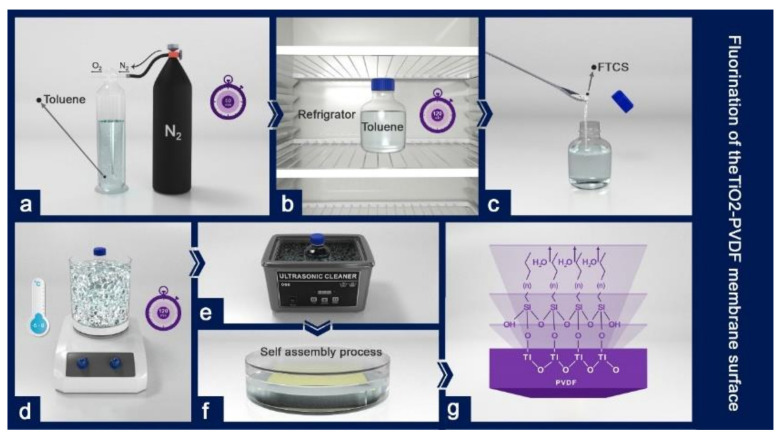
The preparation process of the FTCS–TiO_2_–PVDF membrane: (**a**) purifying toluene using nitrogen; (**b**) reducing the temperature of toluene in a refrigerator; (**c**) mixing toluene with FTCS; (**d**) stirring the mix at a low temperature (−5 °C–0 °C); (**e**) using ultrasonication and ice bath to dissolve the mix completely; (**f**) the self-assembly process of the FTCS solution on the TiO2–PVDF membrane; (**g**) the FTCS–TiO_2_–PVDF membrane.

**Figure 4 polymers-12-02774-f004:**
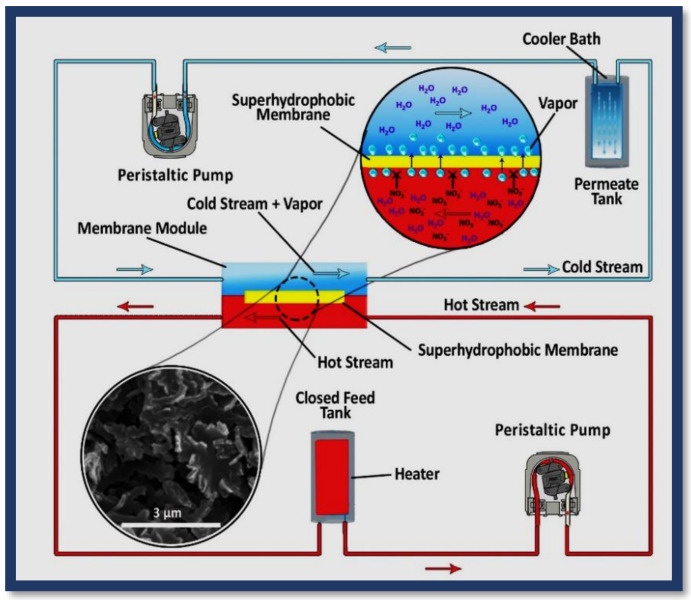
Lab-scale schematic diagram of direct contact membrane distillation setup.

**Figure 5 polymers-12-02774-f005:**
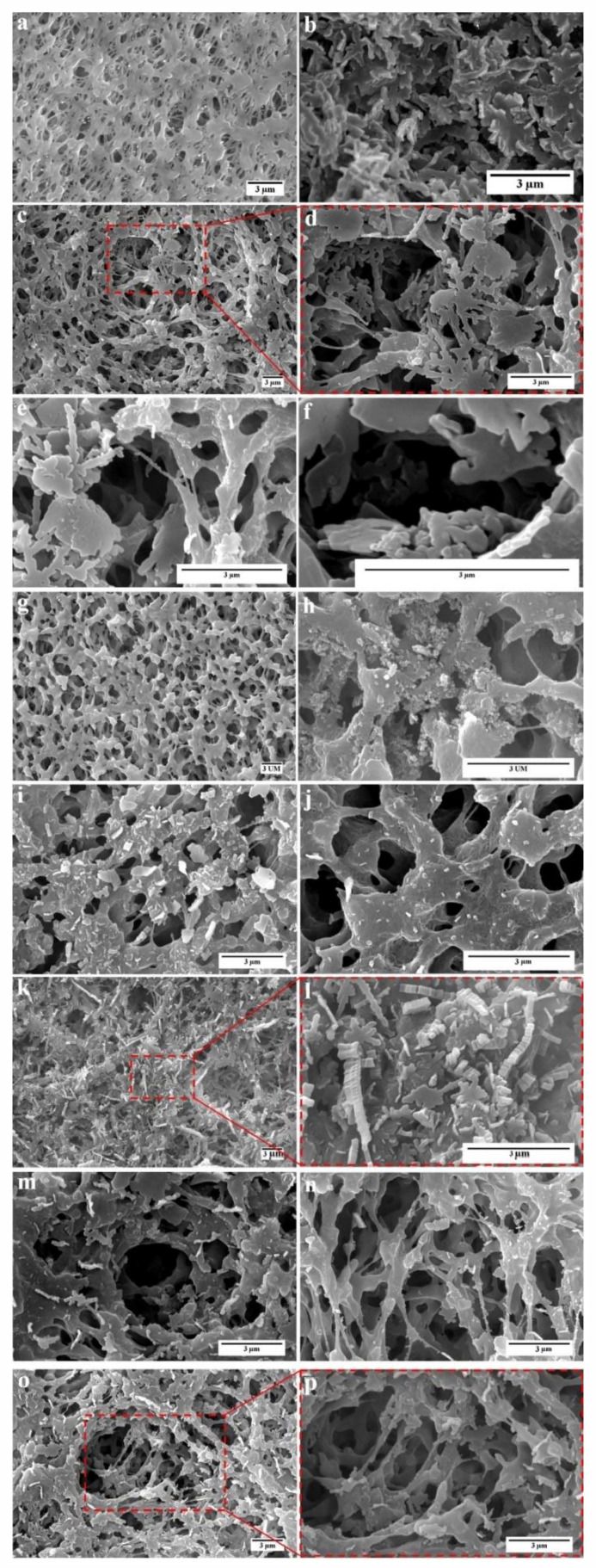
The FESEM images of (**a**) the virgin PVDF membrane; (**b**–**f**) the FTCS–TiO2–PVDF membrane at different magnifications; (**g**,**h**) the TiO2–PVDF membrane in different magnifications; (**i**–**p**) the FTCS–PVDF membrane in different magnifications.

**Figure 6 polymers-12-02774-f006:**
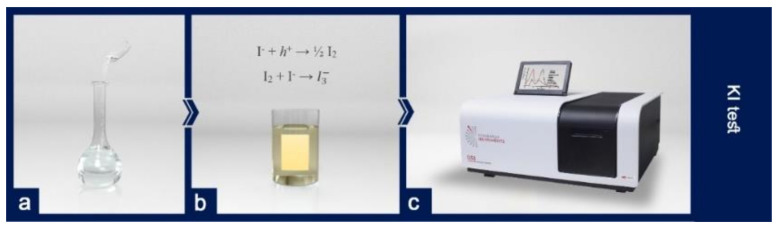
Investigating the durability of TiO_2_ nanoparticles on the surface of the PVDF membrane using the KI test: (**a**) preparing the KI solution; (**b**) the change in the color of the colorless solution by entering the TiO_2_–PVDF membrane; (**c**) the UV-Vis spectrophotometer to show the solution wavelengths.

**Figure 7 polymers-12-02774-f007:**
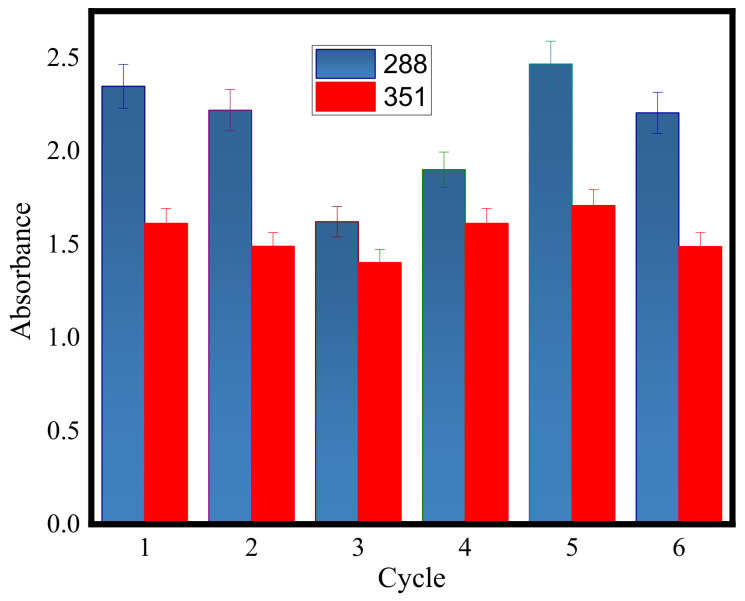
The absorption diagram of the KI solution containing the TiO_2_–PVDF membrane at the wavelengths of 351 and 288 nm after it was exposed to UV for 6 h and for 6 periods.

**Figure 8 polymers-12-02774-f008:**
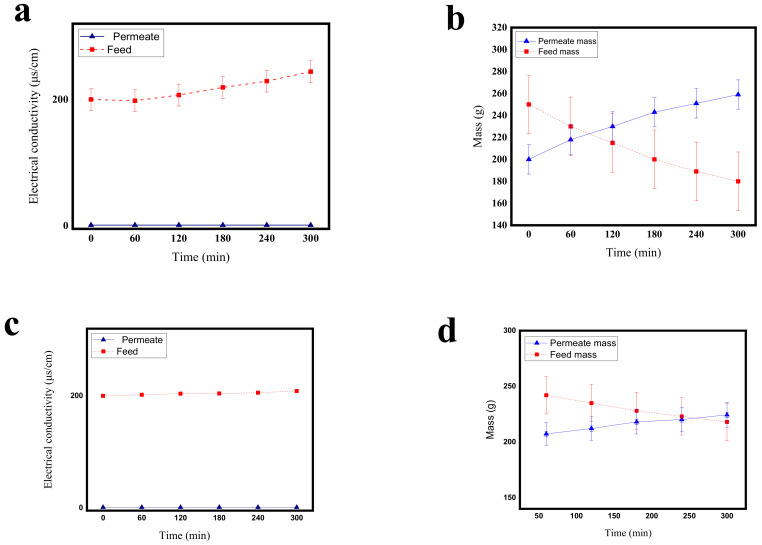
(**a**) The electrical conductivity-time diagram of the membrane distillation process using the PVDF membrane; (**b**) Mass-time diagram of the membrane distillation process using the PVDF membrane; (**c**) Electrical conductivity-time diagram of the membrane distillation process using the FTCS–TiO_2_–PVDF membrane; (**d**) Mass-time diagram of the membrane distillation process using the FTCS–TiO_2_–PVDF membrane.

**Table 1 polymers-12-02774-t001:** Comparing the results of the operational parameters at the pilot and laboratory scales.

Results of Operational Parameters	Lab Scale	Pilot Scale
Feed-permeate temperature (°C)	77–12.30	48.33–21.62
KNO_3_ concentration (g/kg)	0.9	35
Feed-permeate outlet mass flow rate (kg/s)	(3.2–0.56) $\times 10^{-3}$	0.99–1
Cross-membrane flux (Kgm^−2^h^−1^)	2.3	0.96
Effective membrane area (m^2^)	14 × 10^−4^	0.5

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
