# Peer review of "Applying Membrane Distillation for the Recovery of Nitrate from Saline Water Using PVDF Membranes Modified as Superhydrophobic Membranes"

_polymers, 2020, doi:10.3390/polym12122774_

Round 1

Reviewer 1 Report

  1. EDX (Energy-dispersive X-ray spectroscopy) should be done to prove TiO2 was in the structure of the membrane.
  2.  Not only the health impact but also the environmental impact of the nitrate should be discussed in the introduction.

Author Response

Special thanks to Reviewer #2 for his/her comments.

Reviewer 2, comment 1: EDX (Energy-dispersive X-ray spectroscopy) should be done to prove TiO2 was in the structure of the membrane.

Response: Thank you for your comment.

Reviewer 2, comment 2: Not only the health impact but also the environmental impact of the nitrate should be discussed in the introduction.

Response: We appreciate the reviewer comments. This point is added to the manuscript as follow:

Nitrate can also impose serious environment issues. Over supply of nitrate-based fertilizers results in a contamination of soil which makes the farm lands unsuitable for future farming. In addition, the high concentration of nitrate in water resources causes the reduction of its oxygen level damaging aquatic life.

Reviewer 2 Report

The author reported a direct contact membrane distillation device by using modified PVDF membrane. The coating of TiO2 and subsquently FTCS provides higher surface roughness and hydrophobicity. The distillation process and the water flux though the modified membrane was conducted at ambient pressure and ~80 °C.  The preliminary result shows effective ion removal of nitrate as a model ion. The whole manuscript is clearly presented and I recommend its publication on Polymers. However, several minor issues have to be corrected.

  1. Is the exchange chamber or the membrane chamber placed horizontally or vertically? Could the author provide details in the experimental part. Does this influence the cross-membrane water flux?
  2. Please check the molar ratios on line 130 "0.45:1:05:0.5:4.76"
  3. "FTCS–PVDF" also appeared in the manuscript, however, it's not clear that is a new substrate or typo of FTCS–TiO2–PVDF.
  4. In the legend of Figure 5, h appeared twice: "(g and h)", "(h-q)". (Line 234)
  5. In Figure 7, please check the unit of the Y-axis.
  6. On line 298, the author stated an value of zero for the "electrical conductivity". Please also provide the sensitivity of the measurement (conductivity meter) in the experimental part. The conductivity can not be zero even for pure water.
  7. Please remove "Ec" from the Figure 8(a) and (c). It is not easy to understand this abbreviation and the Y-axis already tells the name.

Author Response

Special thanks to Reviewer #1 for his/her comments.

Reviewer 1, comment 1: Is the exchange chamber or the membrane chamber placed horizontally or vertically? Could the author provide details in the experimental part. Does this influence the cross-membrane water flux?

Response: Thank you very much for the point. The membrane test was horizontally performed.

The cross-flow setup is enclosed in the Supplementary Information as follows:

Fig. A.1. Lab scale setup of Direct Contact Membrane Distillation 1) Bain-marie, 2) Thermometer, 3) Membrane module, 4) Cold water inlet - Hot water outlet, 5) Cooler bath, 6) Inlet and outlet of cold water, 7) Conductivity meter, 8) Hot water inlet - Cold water outlet, 9) Peristaltic pump, 10) Cold water, 11) Hot water, 12) Digital Balance

Based on your decision it will be moved to the main context.

Reviewer 1, comment 2: Please check the molar ratios on line 130 "0.45:1:05:0.5:4.76"

Response: The authors greatly appreciate this valuable comment. This section is updated as follows:

  • Membrane Surface Modification

In order to synthesize titanium oxide nanoparticles (Error! Reference source not found.), ethanol, 2,4-pentanedione, perchloric acid 70%, titanium tetraisopropoxide, and Milli-Q water were mixed, at room temperature to form a stable sol of the nanoparticles. The sol was stirred for an hour. The molar ratios of each component in the resulting sol were TTIP: Pluronic F127: 2,4-pentanedione: HClO4:H2O: Ethanol = 1:0.004:0.5:0.5:0.45:4.76. PEG (1000 g/mol) is also substituted with the Pluronic F127 to see the effect of hydrophilic templating agent on the coating films [31].

Reviewer 1, comment 3: "FTCS–PVDF" also appeared in the manuscript, however, it's not clear that is a new substrate or typo of FTCS–TiO2–PVDF.

Response: We appreciate the reviewer's comment.

FTCS-PVDF stands for the membrane substrate which does not have any TiO2 layer whereas FTCS-TiO2-PVDF is the membrane that possesses TiO2 layer that functionalized with FTCS to render the membrane superhydrophobicity.

Reviewer 1, comment 4: In the legend of Figure 5, h appeared twice: "(g and h)", "(h-q)". (Line 234)

Response: Thank you for your valuable comment. This is amended as follows:

Fig. 5. The FESEM images of (a) the virgin PVDF membrane; (b-f) the FTCS–TiO2–PVDF membrane at different magnifications; (g and h) the TiO2–PVDF membrane in different magnifications; (i-q) the FTCS–PVDF membrane in different magnifications.

Reviewer 1, comment 5: 1.   In Figure 7, please check the unit of the Y-axis.

Response: We appreciate the reviewer's critical comment. This is amended as follows:

Fig. 7. The absorption diagram of the KI solution containing the TiO2–PVDF membrane at the wavelengths of 351 nm and 288 nm after it was exposed to UV for 6 hours and for 6 periods.

Reviewer 1, comment 6: 1.   On line 298, the author stated a value of zero for the "electrical conductivity". Please also provide the sensitivity of the measurement (conductivity meter) in the experimental part. The conductivity cannot be zero even for pure water.

Response: We appreciate your comments. The EC value was checked again and found 0.17 uS. The text has been updated.

Reviewer 1, comment 7: Please remove "Ec" from the Figure 8(a) and (c). It is not easy to understand this abbreviation and the Y-axis already tells the name.

Response:

We highly appreciate the reviewer comment. This is amended as follows:

a

b

c

d

Fig. 8. (a) The electrical conductivity-time diagram of the membrane distillation process using the PVDF membrane; (b) Mass-time diagram of the membrane distillation process using the PVDF membrane; (c) Electrical conductivity-time diagram of the membrane distillation process using the FTCS–TiO2–PVDF membrane; (d) Mass-time diagram of the membrane distillation process using the FTCS–TiO2–PVDF membrane.
